# Dynamic Contrastive Learning for Time Series Representation

## Abstract

Understanding events in time series is an important task in a variety of contexts. However, human analysis and labeling are expensive and time-consuming. Therefore, it is advantageous to learn embeddings for moments in time series in an *unsupervised* way, which allows for good performance in classification or detection tasks after later minimal human labeling. In this paper, we propose *dynamic contrastive learning* (DynaCL), an unsupervised contrastive representation learning framework for time series that uses temporal adjacent steps to define positive pairs. DynaCL adopts N-pair loss to dynamically treat all samples in a batch as positive or negative pairs, enabling efficient training and addressing the challenges of complicated sampling of positives. We demonstrate that DynaCL embeds instances from time series into semantically meaningful clusters, which allows superior performance on downstream tasks on a variety of public time series datasets. Our findings also reveal that high scores on unsupervised clustering metrics do not guarantee that the representations are useful in downstream tasks.

## 1 Introduction

A common task in time series (TS) analysis is to split the series into many small windows and identify or label the event taking place in each window. Learning a good representation for these moments eases the time and domain expertise needed for this data annotation. Self-supervised learning, which produces descriptive and intelligible representations in natural language processing (NLP) and computer vision (CV), has emerged as a promising path for learning TS representation. One approach to representation learning is contrastive learning, in which positive and negative pairs of samples are identified, the embeddings of positive pairs are made similar, and the embeddings of negative pairs are made dissimilar. In CV, data augmentation has been successful in creating positive pairs in an unsupervised way. In TS analysis, it is instead possible to create positive pairs on the assumption that moments close in time are also likely to have similar embeddings. Currently, TS representation learning that leverages temporal information in the contrastive objective relies on inefficient sampling positives (Yue et al., 2022; Luo et al., 2023; Oord et al., 2018; Tonekaboni et al., 2021; Woo et al., 2022). This work introduces *dynamic contrastive learning* (DynaCL), an approach to TS representation learning through a simple contrastive learning framework that efficiently captures temporal information by sampling positives from adjacent time steps.

Our contrastive objective extends the N-pair loss introduced by Sohn (2016) to efficiently harness every time step in a sequence as positive and negative pairs. The N-pair loss solves the problem of selecting statistically relevant and varying window sizes in every batch, allowing our method to adapt to the different data structures without prior knowledge of the data distribution. Inspired by the finite difference heat equation in thermodynamics (Mitchell & Griffiths, 1980), we use multiple adjacent moments as positive partners for the reference time step to enhance convergence. Motivated by feature prediction methods (Assran et al., 2023; Caron et al., 2021; Oquab et al., 2023), we extend our model by incorporating a *margin* into the contrastive loss (DynaCL-M) in a bid to introduce feature invariance, and train this variant to jointly optimize both the contrastive and feature prediction objectives. Both of these approaches learn representations before using any human effort on labeling.

As in Tonekaboni et al. (2021), we measure the quality of our learned embeddings using statistics measuring properties of the resulting clusters. However, as the goal is not just to create clusters, but

to learn generalizable features that are useful for downstream classification, we use linear evaluation with a frozen backbone to evaluate the quality of the learned representations. Our findings demonstrate that DynaCL not only produces useful off-the-shelf representations but also outperforms previous TS contrastive learning state-of-the-art methods. This paper makes three main contributions:

- Propose DynaCL, an unsupervised contrastive representation learning framework for time series that samples positive pairs from temporal adjacent steps, and DynaCL-M, an augmentation of DynaCL that combines the contrastive learning objective with a masked feature prediction loss to learning time series representation.

- Introduce multiple positive pairs in the normalized temperature-scaled cross-entropy loss (NT-Xent) Chen et al. (2020) loss to accelerate learning and adapt this for time series representation learning. For convenience, we term this loss *MP-Xent* (multiple positive cross-entropy loss).

- Conduct extensive experiments on three public datasets and demonstrate superior results compared to state-of-the-art baselines on clustering and classification.

## 2 RELATED WORK

**Contrastive representation learning.** Contrastive learning (CL) (Hadsell et al., 2006) is a widely used self-supervised learning strategy with huge success in CV and NLP (Chen et al., 2020; He et al., 2020; Brown, 2020). Unlike generative models that try to reconstruct inputs, contrastive-based methods aim to learn data representation by contrasting positive and negative samples. Sohn (2016) introduces the N-pair loss for efficient learning by employing multiple negatives in each batch update. Specifically, Sohn (2016) extends triple loss (Weinberger et al., 2005) by allowing joint comparison among negative samples. Contrastive predictive coding (CPC) (Oord et al., 2018) learns representation using autoregressive models to predict future time steps in a latent space. A key component of CPC is the introduction of InfoNCE loss, based on noise-contrastive estimation (Gutmann & Hyvärinen, 2010; Jozefowicz et al., 2016) by removing the proximal constraint and using positive pairs. SimCLR (Chen et al., 2020) uses data augmentation and a contrastive loss called NT-Xent that encourages positive pairs (augmented view of the same image) to be closer in the representation space while pushing negative pairs apart. He et al. (2020) proposes a CL framework that uses a momentum encoder to update the features stored in a dynamic dictionary for stable and consistent feature representation over time. Mitrovic et al. (2020) enforces invariance by adding regularization to the InfoNCE objective. Yeh et al. (2022) further removes the positive pair in the denominator, while in Dwibedi et al. (2021), instead of relying solely on augmentations, uses the nearest neighbor of the current data point in feature space to serve as positive pairs. In this work, we extend the NT-Xent loss by introducing multiple positive pairs in the numerator to capture adjacent time steps, we call this modified loss MP-Xent.

**Contrastive learning in time series.** With the recent traction of CL in CV and NLP, several works in TS representation learning have proposed different methods for sampling positive and negative pairs. Wickstrøm et al. (2022) creates a new augmented sample of a time series and attempts to predict the strength of the mixing components. Zhang et al. (2022) samples positive pairs as time-based and frequency-based representations from the time series signal and introduces a time-frequency consistency framework. Yang et al. (2022) introduces dynamic time warping (DTW) data augmentation for creating phase shifts and amplitude changes. Lee et al. (2024) proposes soft assignment to leverage every pair other than the positive pairs by assigning weights to both instance and temporal CL to improve on previous CL frameworks. However, this soft assignment is precomputed offline and not during training. To learn discriminative representation across time, TS2Vec (Yue et al., 2022) considers the representation at the same time stamp from two views as positive pairs. InfoTS (Luo et al., 2023) focuses on developing criteria for selecting good augmentation in contrastive learning in the TS domain. T-loss (Franceschi et al., 2019) employs a time-based sample and a triplet loss to learn representation by selecting positive and negative samples based on their temporal distance from the anchor. TNC (Tonekaboni et al., 2021) presents temporal neighborhood with a statistical test to determine the neighborhood range that it treats as positive samples. Yèche et al. (2021), on the other hand, selects neighbors based on both instance-level and temporal-level criteria with a trade-off parameter allowing the model to balance instance-wise distinction with temporal coherence. (Kiyasseh et al., 2021) define a positive pair as a representation of transformed instances of the same subject. TS-TCC (Eldele et al., 2021) proposes a method to combine temporal

and contextual information in TS using data augmentation to select positives and predict the future of one augmentation using past features of another representation in the temporal contrasting module. CoST (Woo et al., 2022) applied CL in learning representation for TS forecasting by having inductive biases in model architecture to learn disentangled seasonal trends.

**Feature prediction in representation learning.** A growing body of work in TS representation learning has attempted to enforce feature invariance by jointly optimizing instance-wise CL with temporal CL (Yue et al., 2022; Lee et al., 2024). However, we argue that selecting positive pairs and negatives, for instance-wise CL based on distance in the feature space, might lead to suboptimal performance. Ideally, pair selection should be guided by semantic similarity in the learned feature space, rather than raw distance. Self distillation methods have sorted to avoid the need for selecting negatives in their training objectives (Grill et al., 2020; Caron et al., 2021). They rely on encoding two augmented views and mapping one to the other using a predictor. To avoid mode collapse in self-distillation due to the absence of negative as in CL, they update one of the encoder weights with the running exponential moving average (EMA) of the other encoder. Chen & He (2021) show that the EMA was not necessary in practice, even though it led to a small performance boost. Logacjov & Bach (2024) uses the traditional pretext of masked reconstruction to learn feature invariance by a random reconstruction of the masked input of one sensor from another. Masked reconstruction approaches have also produced noteworthy results in forecasting tasks (Dong et al., 2023). TST (Zerveas et al., 2020) attempts to reconstruct masked timestamps using transformers, while PatchTST (Nie et al., 2023) aims to predict subseries of masked patches to learn local invariant features. We adopt the mask approach in our DynaCL-M variant but treat it as a feature prediction task rather than reconstruction, similar to the self-distillation methods. Predicting in representation space has been shown to produce versatile representation with good performance in downstream tasks (Assran et al., 2023; Oquab et al., 2023), as well as eliminate irrelevant data-level details from the target representation.

## 3 PROPOSED ARCHITECTURE: DYNACL

Our main objective is to learn useful representation from instances of time series data. We assume similarity within nearby instances – that consecutive instances in a sequence have the same class and event labels would not change too often. This condition often holds for time series, which have repeated labels in the temporal dimension. DynaCL learns a mapping function $f_\theta : \boldsymbol{x} \to \boldsymbol{z}$, such that given a time series sequence with length $T$, $\boldsymbol{x} = \{x_1, x_2, \ldots, x_T\}$, where $x_i \in \mathbb{R}^{1 \times D}$, projects this series to a representation space $\boldsymbol{z} = \{z_1, z_2, \ldots, z_T\}$, where $z_i \in \mathbb{R}^{1 \times F}$ where $T$ is the sequence length, $D$ is in the input dimension and $F$ is the dimension of the learned embeddings. To that end, we proposed DynaCL and DynaCL-M (Figure 1). To learn from a training sequence $\boldsymbol{x}$ of TS instances in DynaCL, we select an anchor (a single instance), then use adjacent instances as positives and every other sample in the sequence as negatives in the MP-Xent loss. The MP-Xent loss, described more fully in Section 3.1, encourages representations of positive pairs to be similar, and representations of negative pairs to be dissimilar.

For the expanded DynaCL-M model, the architecture consists of an encoder, $E_\theta(.)$, which computes the representation $\boldsymbol{z}$ from the masked input $\boldsymbol{x}^{\mathrm{m}}$, and a linear projector $P_\phi(.)$ that projects the original unmasked input $\boldsymbol{x}$ to a target representation $\bar{\boldsymbol{z}}$ to serve as targets in the feature prediction MSE objective. For the MP-Xent loss, we reused the learned representation $\boldsymbol{z}$, along with the margin, to ensure that the learned representations are pushed further apart.

### 3.1 MULTIPLE POSITIVES CROSS-ENTROPY (MP-XENT)

Sohn (2016) N-pair loss uses every sample in a batch to compute an (N+1) tuple loss. SimCLR (Chen et al., 2020) builds on this by treating augmented views as positive pairs and all other samples in the batch as negatives. In each batch update, every sample serves as a positive pair at least once. We extend this to time series (TS) representation learning by using each instance (1-2 seconds of TS) within a sequence of length T. For a batch of size N and sequence T, we select each time step as an anchor, adjacent steps as positives, and the rest as negatives, forming an NT-tuple loss (Figure 2a). We train our encoder network $E_\theta(.)$ to learn a representation that clusters similar time series while pushing apart dissimilar time series in space using the MP-Xent objective. The encoder $E_\theta(.)$ takes an input $\boldsymbol{x}$ such that $\boldsymbol{z} = E_\theta(\boldsymbol{x})$ where $\boldsymbol{z}$ is the learned feature representation. Given a single

batch $i$, If $z_{i,t}$ and $z_{i,t+1}$ are two consecutive time steps in a sequence of length $T$, with $z_{i,t}$ and $z_{i,t+1} \in \mathbb{R}^{1 \times F}$, equation 1 shows the NT-Xent loss.

$$\ell(i,t) = -\log \frac{\exp(\text{sim}(z_{i,t}, z_{i,t+1})/\tau)}{\sum_{k=1}^{T} \mathbf{1}_{[k \neq t]} \exp(\text{sim}(z_{i,t}, z_{i,k+1}/\tau)} \tag{1}$$

Where $T$ is the sequence length, $\tau$ is the temperature parameter (Chen et al., 2020), and cosine similarity is the similarity score. Implementing the objective in equation 1 leads to slower convergence. Building upon the principles of the finite difference method (Mitchell & Griffiths, 1980), we extend the NT-Xent loss objective in equation 1 to account for multiple positives for faster convergence and efficient training, as shown in Figure 2b. As before, given a reference time step $z_{i,t}$ with adjacent time steps $z_{i,t-1}$ and $z_{i,t+1}$, our MP-Xent loss is as follows.

$$\ell(i,t) = -\log \frac{\exp(\text{sim}(z_{i,t}, z_{i,t-1})/\tau) + \exp(\text{sim}(z_{i,t}, z_{i,t+1})/\tau)}{\sum_{k=1}^{T} \mathbf{1}_{[k \neq t, t-1, t+1]} \exp(\text{sim}(z_{i,t}, z_{i,k+1})/\tau) + \sum_{l=1}^{T} \mathbf{1}_{[l \neq t, t-1]} \exp(\text{sim}(z_{i,t-1}, z_{i,l})/\tau)} \tag{2}$$

For the entire sequence length $T$ and batch $N$, we have an *NT* tuple loss per update, making our training very efficient.

$$L_{MP-Xent} = \frac{1}{NT} \sum_{i=1}^{N} \sum_{t=1}^{T} \ell(i,t) \tag{3}$$

In our DynaCL model, we focus on optimizing only equation 3.

## 3.2 DYNAMIC CONTRASTIVE LEARNING WITH MARGIN (DYNACL-M)

In this section, we introduce a variant called DynaCL-M. DynaCL-M contains two augmentations to DynaCL. First, we add feature prediction. Second, we add a dynamic margin to further separate dissimilar but adjacent time steps. We will use comparisons between DynaCL and DynaCL-M to demonstrate the lack of correlation between clustering metrics and downstream effectiveness in Section 4. In that same section, we will perform an ablation study to illustrate the impact of each augmentation.

Feature prediction has been shown to learn invariant representations by guiding the model to focus on relevant underlying patterns rather than high-level details (Assran et al., 2023; Oquab et al., 2023). To enforce feature invariance in our learned representation for highly dynamic datasets, we extend DynaCL by introducing feature prediction into our MP-Xent objective.

We mask our input $x$ to give $x^{\text{m}}$, then compute the representation $z$ from the masked input $x^{\text{m}}$. Additionally, we project the unmasked $x$ to a target vector, $\bar{z} = P_\phi(x)$, where $P_\phi$ is a randomly-chosen projection operator into $F$ dimensions, making $\bar{z}$ the same size as $z$. We encourage the encoding of the masked $x^{\text{m}}$ to be close to the projection of the unmasked $x$ using the $L_{MSE}$ loss for mask feature prediction.

$$L_{MSE} = \frac{1}{NT} \sum_{i=1}^{N} \sum_{t=1}^{T} \|z_{i,t} - \bar{z}_{i,t}\|^2 \tag{4}$$

In addition, in this variant, we introduce dynamic margins in the similarity vectors that increase the distance between features in the representation space if two adjacent time steps are dissimilar based on a threshold hyperparameter. Given an input $x$ with sequence length $T$, we precompute a pseudo-label $\mathcal{Y}$ based on the cosine similarities between consecutive time steps and threshold as

$$\mathcal{Y} = \begin{cases} 0 & \text{if sim}(z_i, z_j) > \text{threshold} \\ 1 & \text{otherwise} \end{cases}$$

We apply this pseudo-label $\mathcal{Y}$ and a constant margin to our similarity matrix $\boldsymbol{M} \in \mathbb{R}^{T \times T}$. This matrix contains the similarity score for all time steps in a single batch. To be specific, element $M_{t,t+1}$ corresponds to $\mathrm{sim}(z_t, z_{t+1})$.

$$\boldsymbol{M}_{\mathrm{margin}} = \frac{1}{2}(1 - \mathcal{Y})\boldsymbol{M}^2 + \frac{1}{2}\mathcal{Y}\left[\max(0, \mathrm{margin} - \boldsymbol{M})\right]^2 \tag{5}$$

**Combine objective for DynaCL-M variant.** The final objective of DynaCL-M combines the MP-Xent and MSE loss using a $\lambda$ hyperparameter. We use the learned representation $z$ both as the target for the MSE and as the input to our MP-Xent loss, as shown in Figure 1. The first term ensures the learned representations have temporal coherence, while the second term enforces feature invariance and training stability through a masked feature prediction.

$$L_{DynaCL-M} = \lambda L_{MP-Xent} + (1 - \lambda)L_{MSE} \tag{6}$$

Here, $\lambda$ is a fixed scalar hyperparameter that represents the relative contribution of each loss term.

---

**Algorithm 1** : DynaCL-M

1: **Input:** Data batch $\{x_i\}_{i=1}^N$, margin $m$, pseudo-labels $\{\mathcal{Y}_i\}_{i=1}^N$, $\lambda$
2: **Initialize:** Model parameters $\theta$
3: **for** batch $n = 1$ to $N$ **do**
4:     **for** sequence $t = 1$ to $T$ **do**
5:         Mask $x_{n,t}$ to get $x_{n,t}^m$
6:         Encode $x_{n,t}^m \rightarrow z_{n,t}$
7:     **end for**
8:     Compute MP-Xent loss using $\bar{z}_n$, $m$, $\mathcal{Y}_n$
9:     **for** sequence $t = 1$ to $T$ **do**
10:        Project $x_{n,t} \rightarrow \bar{z}_{n,t}$
11:        Compute MSE loss between $z_{n,t}$ and $\bar{z}_{n,t}$
12:     **end for**
13:     Jointly optimize MP-Xent and MSE losses using $\lambda$
14: **end for**
15: **Output:** Optimized $\theta$

---

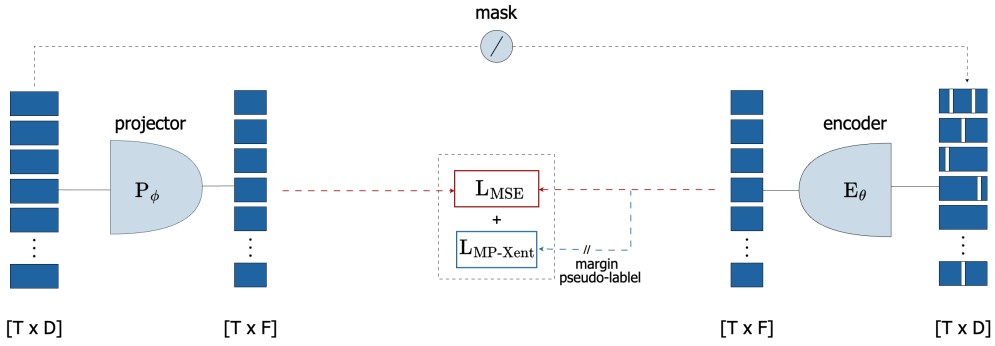

Figure 1: Unsupervised representation learning using dynamic contrastive learning with margin (DynaCL-M). We train on TS instances of length T and feature dimension D. (Right to left): We mask random features from the time series instances and use this as input to the encoder. The encoder processes this mask input to generate an embedding vector.

### 3.3 NETWORK ARCHITECTURE

We use a simple convolutional neural network (CNN) architecture as our feature extractor backbone in our encoder $E_\theta(\cdot)$. The CNN network consists of 3 blocks of 1D convolution with a kernel size of

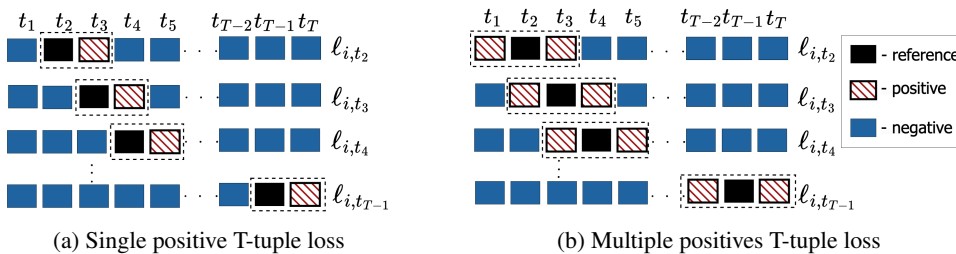

(a) Single positive T-tuple loss        (b) Multiple positives T-tuple loss

Figure 2: Positive pairs selection for the contrastive learning objective. The index $i$ refers to the current batch (a) The (N+1) tuple loss (Sohn, 2016) operates on batch N, we adapted this to TS of sequence length T and batch N to obtain NT-tuple losses per batch. (b) We further extend the NT-Xent loss (Chen et al., 2020) by introducing multiple positives for a given reference step from adjacent time steps to enhance convergence. We skip the reference instance $t_1$ and $t_T$ as both losses behave similarly at the edges.

1, followed by batch normalization and ReLU activation, with an embedding dimension of 32. Since our focus is on developing a loss function, we use the same architecture for all baselines in Section 4. To preprocess the time series signal for our encoder, we perform a short-time Fourier transform (STFT) on the signal to obtain input with dimension $B \times T \times D$, where $B$ is the batch size, $T$ is the sequence length, and $D$ is the input dimension (see Appendix D for more details on data processing for each dataset). We apply a mask to a random fraction of features from the input $x$ by assigning these values to 0. The encoder $E_\theta$ processes the masked input $x^m$ to predict a feature representation $z$. For the projector network $P_\phi$, we use a single-layer linear network to project the unmasked input $x$ to a 32-dimensional vector $\bar{z}$, which serves as the target in the MSE loss (see Figure 1).

## 4 EXPERIMENTS

In this section, we evaluate the performance of our proposed method on three benchmark datasets to assess the quality of the learned embeddings. We compare our approach against state-of-the-art baselines for time series representation learning on clustering quality and event classification using linear fine-tuning with a frozen backbone. Additionally, we qualitatively compare learned embeddings alongside those from previous methods using t-SNE plots in Figure 3. These experiments demonstrate that DynaCL-M builds more compact and separate clusters than other methods, but that vanilla DynaCL outperforms other approaches in building semantically meaningful representations. We perform an ablation study to highlight the effects of the different components of our DynaCL models. Lastly, though it is not a focus of our work, we show that the simplicity of DynaCL allows it to be trained the fastest of all tested methods.

We evaluate our model on three public datasets on human activity recognition, electrical activity of the heart, and sleep stage classification. Table 1 shows the summary statistics of these datasets.

Table 1: Summary of dataset distributions used across all experiments. An instance is a preprocessed block of TS. For the HARTH and ECG, each instance is 1 second while for the SLEEPEEG an instance is 2 seconds.

|  | # Instance | Sequence length | Dimension | Classes | Frequency (Hz) |
|---|---|---|---|---|---|
| **HARTH** | 1,270,087 | 119 | 156 | 12 | 50 |
| **SLEEPEEG** | 371,055 | 300 | 178 | 5 | 100 |
| **ECG** | 1,531,771 | 119 | 252 | 4 | 250 |

**HARTH** - This is a human activity recognition (HAR) dataset (Logacjov et al., 2021) that contains recordings from 22 participants, each wearing two 3-axial Axivity AX3 accelerometers for approximately 2 hours in a free-living setting at a sampling rate of 50Hz. This dataset comprises 12 distinct classes of varying human activities *(standing, lying, walking, shuffling, running, sitting, stairs - ascending and descending, and four different cycling positions)*. We preprocess the signal by applying a short-time Fourier transform (STFT) using a one-second Hann window (Blackman & Tukey, 1958;

Logacjov & Bach, 2024) with a half-second overlap. We then concatenate the activities from all 22 subjects to build a continuous time series, resulting in a spectrogram with 1,270,087 instances and 156 feature dimensions. During our unsupervised representation learning, for each iteration, we use a sequence length of 119 instances, corresponding to 60 seconds, and encode the representations in a 32-dimensional space.

SLEEPEEG - This dataset (Goldberger et al., 2000) contains 153 whole-night electroencephalography (EEG) sleep recordings from 82 healthy subjects, sampled at 100 Hz. We use the preprocessed dataset from Zhang et al. (2022), which is segmented with a window size of 200, resulting in 371,055 instances with a feature dimension of 178. Each sample corresponds to one of the five sleep stages: Wake (W), Non-Rapid Eye Movement (N1, N2, N3), and Rapid Eye Movement (REM). In our training, we use a sequence length of 300 and output representations in a 32-dimensional space.

ECG - We use the MIT-BIH Atrial Fibrillation dataset (Moody, 1983), which includes 25 long-term electrocardiogram (ECG) recordings of human subjects with atrial fibrillation, each with a duration of 10 hours. The dataset contains two ECG signals, each sampled at 250 Hz, with annotations marking the different rhythms: atrial fibrillation (A), atrial flutter (F), AV junctional rhythm (AV), and all other rhythms. Similar to the HARTH dataset, we apply a short-time Fourier transform (STFT) with a one-second Hann window and a half-second overlap, producing a total of 1,531,771 instances with a feature dimension of 252. Finally, we select 119 instances, corresponding to 60 seconds as sequence length. The learned representations are encoded in a 32-dimensional space. This dataset is particularly useful for evaluating how our proposed method performs on imbalanced data, as the atrial (A) rhythm and "all other rhythms" account for more than 99% of the entire dataset.

We compare our model with five state-of-the-art approaches in time series representation learning: InfoTS (Luo et al., 2023), CPC (Oord et al., 2018), TNC (Tonekaboni et al., 2021), TS2Vec (Yue et al., 2022) and CoST (Woo et al., 2022). InfoTS maximizes agreement between representations of the same subseries through temporal augmentations. CPC extracts useful representations by predicting future latent representations in a sequence. TNC learns by contrasting data points within the same neighborhood against those from different neighborhoods. TS2Vec captures both global and temporal dependencies by contrasting time series across different scales and timesteps, while CoST employs a two-step approach to TS forecasting by learning disentangled seasonal trends. To ensure a fair comparison, all models were trained using the same preprocessing pipeline and hyperparameters. Specifically, we employed the AdamW optimizer with a learning rate of $1e^{-3}$ and a batch size of 8. Additionally, to eliminate any performance differences arising from variations in model architecture, we use the same encoder network across all baselines. We aim to compare the learning frameworks independent of the choice of encoder. To this end, we selected a simple CNN architecture to assess how effectively each framework can leverage the limited capacity of a basic encoder to learn meaningful representations. Consequently, we substituted the dilated CNN unique to the TS2Vec encoder with a regular 1D CNN. All experiments were conducted on an NVIDIA Tesla V100 GPU (refer to Appendix B for more details on each baseline and implementation).

## 4.1 CLUSTERABILITY

Even though our final objective is to learn useful embeddings for downstream tasks, we echo the evaluation in Tonekaboni et al. (2021) by checking the properties of the distribution of the representations in the encoding space. Bengio et al. (2014) posit that the formation of natural clustering is one of the properties of a good representation. To capture the performance of each baseline on clustering, we use two popular clustering evaluation metrics, namely Davies-Bouldin Index (DBI) (Davies & Bouldin, 1979) and Silhouette Score (SS) (Rousseeuw, 1987). DBI measures the average similarity ratio of each cluster with its most similar cluster. A lower DBI score indicates better separation between clusters. SS evaluates how similar an object is to its own cluster compared to other clusters. SS values range from -1 to 1, with higher values reflecting both compactness and separation. Table 2 shows the result of our approach against the baseline methods on these unsupervised clustering measures. Overall, our proposed DynaCL-M outperforms all four baselines in two out of the three datasets, and performs competitively on the third, the highly imbalanced ECG dataset. While strong cluster scores are not our ultimate goal, DynaCL-M performs well in the metrics used in other papers in this field.

Table 2: Comparison with state-of-the-art methods in clustering. All models are evaluated on test sets that were not used during pretraining. DynaCL-M demonstrates superior clustering performance on two of the three datasets and ranks second to TNC (Tonekaboni et al., 2021) on the ECG.

| | HARTH | | SLEEPEEG | | ECG | |
|---|---|---|---|---|---|---|
| | DBI↓ | Silhouette ↑ | DBI↓ | Silhouette ↑ | DBI↓ | Silhouette↑ |
| CoST | 1.46±0.05 | 0.25±0.03 | 2.13±0.18 | 0.28±0.02 | 1.15±0.10 | 0.45±0.01 |
| CPC | 1.65±0.11 | 0.17±0.02 | 2.59±0.10 | 0.26±0.01 | 1.88±0.06 | 0.16±0.02 |
| TNC | 0.60±0.11 | 0.67±0.24 | 0.58±0.07 | 0.21±0.00 | **0.48±0.06** | **0.96±0.02** |
| InfoTS | 0.96±0.08 | 0.58±0.03 | 0.67±0.08 | 0.24±0.01 | 0.97±0.10 | 0.63±0.03 |
| TS2Vec | 1.23±0.07 | 0.59±0.02 | 1.01±0.05 | 0.31±0.01 | 0.74±0.03 | 0.64±0.02 |
| DynaCL | 1.07±0.06 | 0.35±0.02 | 1.12±0.25 | 0.41±0.20 | 0.98±0.23 | 0.63±0.13 |
| DynaCL-M | **0.46±0.01** | **0.95±0.01** | **0.48±0.01** | **0.87±0.05** | 0.51±0.11 | 0.72±0.12 |

## 4.2 LINEAR EVALUATION WITH FROZEN BACKBONE

Our main goal is to learn representations that are useful in downstream classification. With that in mind, we train a linear classifier on top of the learned representations to assess how well the learned features generalize to the task of interest when used by a simple classifier, which is reflective of real-world usage where the learned representations are further fine-tuned or used for downstream tasks. We fine-tuned a linear classifier with a frozen backbone on the features from the learned representation and evaluated the performance of our model on the test set. We perform an 80-20 subject-wise train-test split. We train our unsupervised models on the 80% data. We then reused this 80% to fine-tune a linear model with a frozen encoder network and evaluate on the remaining 20%. We have presented our results on the accuracy, F1 score, precision, and recall metrics in Table 3.

Table 3: Comparison with state-of-the-art methods on linear evaluation with a frozen backbone. We compare DynaCL with state-of-the-art baselines and a randomly initialized encoder (*Random Init.*) on frozen evaluation. We train a linear classifier on top of the from an encoder on the 80% train set (excluding the *all other rhythms* for the ECG dataset) and evaluate the remaining 20%. We train 5 different runs for 50 epochs on all datasets. DynaCL achieved the best performance on all three datasets.

| Datasets | Models | Accuracy | F1 score | Precision | Recall |
|---|---|---|---|---|---|
| HARTH | Random Init. | 34.45±4.01 | 0.26±0.03 | 0.18±0.02 | **0.14±0.01** |
| | CoST | 32.74±9.67 | 0.26±0.06 | 0.17±0.02 | 0.15±0.02 |
| | CPC | 27.80±4.65 | 0.20±0.03 | 0.14±0.01 | 0.12±0.01 |
| | TNC | 30.14±1.04 | 0.16±0.02 | 0.06±0.02 | 0.10±0.01 |
| | InfoTS | 33.73±2.48 | 0.21±0.03 | 0.16±0.03 | 0.12±0.02 |
| | TS2Vec | 35.58±1.51 | 0.24±0.02 | 0.15±0.02 | 0.12±0.01 |
| | DynaCL | **37.95±4.51** | **0.29±0.06** | **0.18±0.04** | 0.13±0.02 |
| | DynaCL-M | 31.31±2.73 | 0.18±0.05 | 0.08±0.05 | 0.10±0.01 |
| SLEEPEEG | Random Init. | 44.94±0.19 | 0.32±0.01 | 0.36±0.01 | 0.26±0.00 |
| | CoST | 50.30±0.23 | 0.39±0.01 | 0.32±0.00 | 0.26±0.01 |
| | CPC | 44.19±0.63 | 0.34±0.02 | 0.35±0.01 | 0.26±0.01 |
| | TNC | 41.78±0.20 | 0.26±0.01 | 0.26±0.01 | 0.21±0.00 |
| | InfoTS | 44.02±0.25 | 0.34±0.01 | 0.38±0.02 | 0.24±0.01 |
| | TS2Vec | 48.43±0.49 | 0.42±0.01 | 0.42±0.02 | 0.31±0.01 |
| | DynaCL | **62.08±0.64** | **0.60±0.01** | **0.52±0.01** | **0.50±0.01** |
| | DynaCL-M | 41.40±4.00 | 0.28±0.01 | 0.24±0.01 | 0.20±0.00 |
| ECG | Random Init. | 55.01±0.00 | 0.51±0.00 | 0.30±0.00 | 0.28±0.00 |
| | CoST | 55.82±4.95 | 0.50±0.08 | 0.31±0.03 | 0.29±0.02 |
| | CPC | 55.36±1.43 | 0.50±0.04 | 0.31±0.00 | 0.29±0.00 |
| | TNC | 47.76±0.02 | 0.31±0.00 | 0.24±0.01 | 0.25±0.00 |
| | InfoTS | 53.46±0.59 | 0.45±0.01 | 0.32±0.01 | 0.28±0.00 |
| | TS2Vec | 49.96±0.65 | 0.41±0.01 | 0.27±0.00 | 0.26±0.00 |
| | DynaCL | **58.74±0.62** | **0.56±0.01** | **0.32±0.00** | **0.30±0.00** |
| | DynaCL-M | 50.05±0.42 | 0.37±0.01 | 0.31±0.00 | 0.26±0.00 |

From Table 3, we see that our DynaCL model exhibits remarkable performance and outperforms all baselines on all three public datasets, despite poor clustering scores. Conversely, DynaCL-M struggles in downstream linear evaluation, despite having better clustering scores. The general lower

scores across all models on the HARTH are due to this dataset having varying activities with more classes than SLEEPEEG and ECG, thereby yielding representations that are not as linearly separable.

## 4.3 VISUALIZATION OF LEARNED REPRESENTATIONS

In addition to our representation being useful in downstream tasks, we also want to learn compact and semantically meaningful representations. We seek to understand how consistently the learned representation clusters similar instances together, despite not having access to this information during training. This is a good indicator of whether the representations are meaningful. To that end, we visualize a random subset from that test set that was not used during training. Figure 3 shows a t-SNE plot of the learned representation from all baselines on all three datasets. Interestingly, despite having lower scores on the unsupervised clustering metrics, our vanilla DynaCL model, compared to other baselines on TS representation learning, seems to embed instances into well-defined, semantically meaningful clusters, challenging the assumption that good scores on these unsupervised clusters necessarily lead to meaningful representations.

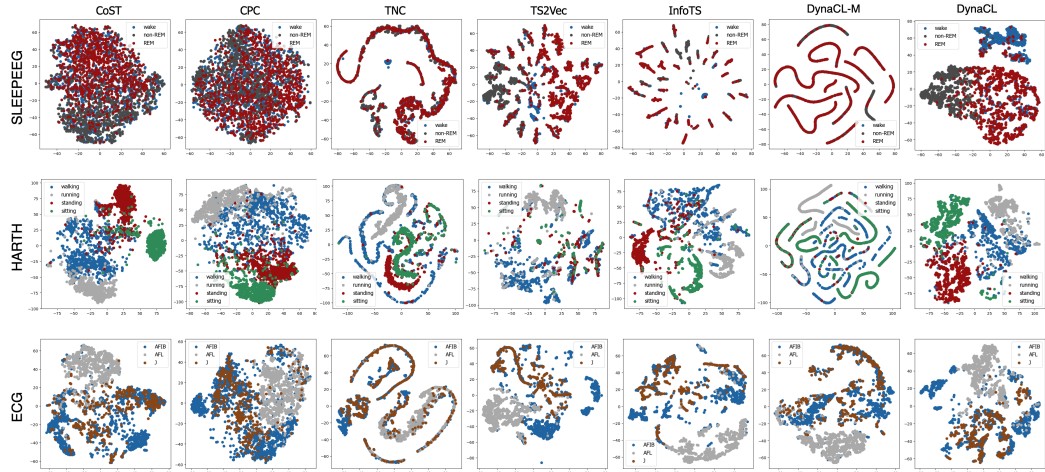

Figure 3: t-SNE visualization of the learned embeddings on random instances on the SLEEPEEG (first row), HARTH (second row), and ECG (third row) test sets across all methods. For the SLEEPEEG each instance (data point) spans 2 seconds, while for the HARTH and ECG each instance is 1 second.

## 4.4 ABLATION STUDY

To investigate the relevance of the individual components of our proposed DynaCL and DynaCL-M methods, we conducted an ablation study. We compare these components on clustering and linear fine-tuning with a frozen backbone. In particular, we check the effect of adding margin and MSE feature prediction loss to the vanilla MP-Xent objective.

From Table 4, we observe that our best-performing model for downstream tasks is DynaCL (MP-Xent only). DynaCL-M (MP-Xent + MSE + Margin), however, achieved better clustering performance and learned useful representation of highly dynamic datasets like the HARTH. We also observe that naively adding the margin in our MP-Xent loss causes the representations to collapse to 0 in all features, making cluster metrics impossible; naturally, this also resulted in lower scores on the downstream tasks. This highlights the significance of the MSE feature prediction term in the combined loss in equation 6, contributing to both the learning of feature invariance and the stability of the training process. Finally, as shown in Table 4, although DynaCL-M achieves the best clustering performance, it struggles in downstream evaluations. This indicates that a successful clustering score does not necessarily result in well-separated or semantically meaningful embeddings. Another notable observation from Figure 4 is that, on the ECG dataset, the performance of the MP-Xent + MSE + margin configuration is identical to that of MP-Xent + MSE, indicating that the inclusion of the margin component did not produce any discernible effect in this case. It is worth noting that

Table 4: Ablation study to understand the impact of different components of our model. We notice that only the DynaCL and DynaCL-M variants stand out across all metrics. Clearly, DynaCL-M produces top scores on unsupervised clustering, while DynaCL shows outstanding performance on downstream evaluation.

| | HARTH | | SLEEPEEG | | ECG | |
|---|---|---|---|---|---|---|
| | DBI↓ | Silhouette ↑ | DBI↓ | Silhouette ↑ | DBI↓ | Silhouette ↑ |
| MP-Xent only (DynaCL) | 1.07±0.06 | 0.35±0.02 | 1.12±0.25 | 0.41±0.20 | 0.98±0.23 | 0.63±0.13 |
| MP-Xent + MSE | 1.05±0.08 | 0.34±0.03 | 1.31±0.06 | 0.24±0.01 | **0.51±0.11** | **0.72±0.12** |
| MP-Xent + margin | - | - | - | - | - | - |
| MP-Xent + MSE + margin (DynaCL-M) | **0.46±0.01** | **0.95±0.01** | **0.48±0.01** | **0.87±0.05** | **0.51±0.11** | **0.72±0.12** |

| | Linear Acc. | F1 Score | Linear Acc. | F1 Score | Linear Acc. | F1 Score |
|---|---|---|---|---|---|---|
| MP-Xent only (DynaCL) | **37.95±4.51** | **0.29±0.06** | **62.08±0.64** | **0.60±0.01** | **58.74±0.62** | **0.56±0.01** |
| MP-Xent + MSE | 36.27±2.55 | 0.26±0.04 | 47.12±0.68 | 0.39±0.01 | 50.05±0.42 | 0.37±0.01 |
| MP-Xent + margin | 28.15±0.00 | 0.12±0.00 | 41.29±0.01 | 0.24±0.00 | 47.76±0.00 | 0.31±0.00 |
| MP-Xent + MSE + margin (DynaCL-M) | 31.31±2.73 | 0.18±0.05 | 41.40±0.28 | 0.28±0.01 | 50.05±0.42 | 0.37±0.01 |

this dataset is also the only one where DynaCL-M was outperformed by the TNC baseline on the unsupervised clustering metrics in Table 2.

## 4.5 TRAINING TIME

The simplicity of DynaCL allows it to train very quickly. In table 5, we show it trains the fastest of all tested methods.

Table 5: Unsupervised pretraining time (in seconds) for all baseline models over 500 epochs on all three datasets.

| | TIME (S) | | |
|---|---|---|---|
| | HARTH | SLEEPEEG | ECG |
| CoST | 13.4k | 1.5k | 3.5k |
| CPC | 4.1k | 0.8k | 2.1k |
| TNC | 6.5k | 1.0k | 2.6k |
| InfoTS | 6.1k | 1.3k | 3.4k |
| TS2Vec | 13.9k | 2.3k | 5.5k |
| DynaCL | **3.4k** | **0.6k** | **1.9k** |
| DynaCL-M | 4.6k | 1.2k | 4.4k |

## 5 CONCLUSION

In this work, we present DynaCL, a method for unsupervised representation learning of time series data. The DynaCL method demonstrates the ability to learn semantically meaningful representations off the shelf and outperforms previous time series representation learning methods in downstream linear evaluation. Additionally, we show that including margin in our MP-Xent objective and jointly optimizing with MSE loss is particularly effective in producing clusters with top scores on the DBI and SS clustering metrics. Our findings, however, indicate that achieving high scores on unsupervised clustering metrics does not necessarily imply that the learned embeddings are meaningful or effective in downstream tasks. Finally, we studied the contribution of individual components of DynaCL and DynaCL-M. We concluded that our best-performing model for downstream tasks is the vanilla DynaCL without the MSE loss and margin, which proves that with a simple positive sampling strategy of selecting adjacent time steps as positive in an NT tuple loss, Our model competes with previous approaches that rely on statistical methods and prediction sampling, where window sizes are selected based on prior knowledge (Tonekaboni et al., 2021; Oord et al., 2018) as well as the use of temporal augmentations (Luo et al., 2023; Yue et al., 2022). Also, DynaCL model not only delivers exceptional performance in downstream classification tasks but also exhibits the shortest training time (Figure 5). This highlights the efficiency of the multiple positive sampling strategy in our MP-Xent contrastive objective.

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

## A    FULL DESCRIPTION OF MODEL ARCHITECTURE

The encoder model is designed to extract features from time-series data using 1D convolutional layers. It reduces the dimensionality of the input while retaining meaningful temporal information. The model consists of three 1D convolutional layers, each followed by batch normalization and ReLU activation to introduce non-linearity and stabilize the training process.

The input to the model is a 3D tensor of shape (batch size, sequence length, input dimension), where the batch size is 8 for all datasets while the sequence length and dimension are dependent on the dataset. After passing through the three convolutional layers, the output dimensionality is reduced to embedding dimension = 32. Each convolutional layer applies a kernel size of 1 to focus on individual time steps, progressively reducing the number of channels from the input dimension to 128, 64, and 32. We apply batch normalization after each convolution to stabilize the activations, and ReLU activation functions introduce non-linearity, ensuring only positive values are passed through. The output tensor is reshaped back to the original order, returning a feature representation of shape (batch size, sequence length, embedding dimension). This architecture efficiently captures temporal dependencies while reducing the dimensionality, making it suitable for downstream tasks such as the classification and prediction of time-series data.

## B    IMPLEMENTATION OF BASELINE MODELS

In this section, we provide the reproduction details for the methods compaired against. All results presented in this work are based on reproduction using code provided by the authors.

**InfoTS** (Luo et al., 2023). We use the code and default parameters provided by the authors for the baseline. Specifically, we set the probabilities of the two different augmentation views as $p = 0.2$, maximum train length = 500, and then the temperature used in contrastive loss functions $\tau_0$ and $\tau_1$ as 2.0 and 0.1, respectively. In the loss function, k=8 is used to define the number of local negatives for the local infoNCE loss function (again, default parameters by authors). Finally, we combine both the global and local infoNCE losses.

**TS2Vec** (Yue et al., 2022). We use the implementation and default parameters provided by the authors for the TS2Vec model. Specifically, we set the maximum sequence length during training to 500. The cropping is performed by selecting two random temporal windows within the sequence, defined by crop lengths and offsets dynamically generated during training. In each epoch, two augmented views of the input sequence are created: $x_1$ and $x_2$, where the lengths of the crops vary slightly. To ensure matching dimensions for the contrastive loss, padding is applied to equalize the output dimensions if one crop is shorter than the other. Finally, the hierarchical contrastive loss is computed based on these two views. We substituted the dilated CNN with our simple 1D CNN encoder to create a fair comparison across all baselines.

**CPC** (Oord et al., 2018). The CPC method has two extra network architectures: the density estimator, which is a linear model, and the auto regressor with a gated recurrent unit (GRU). We select encodings from the middle of the sequence and with a window of size 5 to select the next 5 future instances. During training, the model uses a contrastive loss based on density ratios derived from the encoded time series instances. After processing the entire sequence through the encoder, a central segment is extracted and passed through a GRU to obtain the context vector $c_t$ This vector is projected using the linear estimator to compute density ratios that measure similarity between the encodings and the projected vector. Negative samples are randomly selected from the encodings, avoiding indices near the center, while the positive sample corresponds to the encoding immediately after the center. These density ratios are concatenated into tensor $X_N$, from which the cross-entropy loss is calculated against a label tensor indicating the positive sample's index (code adapted from the authors).

**TNC** (Tonekaboni et al., 2021). For the TNC, we adopt all relevant functions from the author code repository, namely: find neighbors, find non-neighbors, and binary cross entropy (BCE) loss function. The authors use a discriminator network to distinguish between two inputs, $x$ and $\bar{x}$, based on their similarity. The model architecture comprises two linear layers with a ReLU activation and dropout for regularization. Specifically, it concatenates the feature vectors of the two inputs into a single tensor, then fed through the model to output a probability score indicating whether the

inputs belong to the same neighborhood. The weights of the linear layers are initialized using the Xavier uniform distribution. We use a Monte Carlo sample size and window size of 20, and w (hyperparameter to control the contribution of the different losses) as 0.1. All hyperparameters are used as provided by the authors and kept the same for all datasets.

**CoST** Woo et al. (2022). For this reproduction of the CoST baseline, we use the implementation and default parameters provided by the authors. The CoST method adapted the Dilated CNN from TS2Vec. To ensure all methods have the same backbone feature extractor we replace this with our 1D CNN used across all methods. The parameters used for this experiment are: kernels = [1, 2, 4, 8, 16, 32, 64, 128], depth = 10, alpha = 0.05, K = 256, sigma = 0.5 and multiplier = 5.

## C UNSUPERVISED PRE-TRAINING SETUP

For the pretraining of all models, we maintain the same parameters for all baselines. For our DynaCL model, we use temperature $\tau = 0.5$. For DynaCL-M, we use temperature $\tau = 0.5$, margin = 5, $\lambda = 1$, mask fraction of 0.3, and threshold of 0.4. For the HARTH dataset, however, we find that mask fraction = 1e-5 (almost no masking) and $\lambda$ of 1e-30 produced the best results. We perform an 80-20 subject-wise train-test split with the total instances for each category shown in Figure 6.

Table 6: Dataset distributions used across all experiments. We pre-train all models for 500 epochs on 80% of the entire data instances and evaluate downstream performance on the remaining 20%. For the HARTH and ECG, each instance is 1 second while for the SLEEPEEG an instance is 2 seconds.

|  | # Train instances | # Test instances | Dimensions | Classes |
|---|---|---|---|---|
| **HARTH** | 1,016,141 | 253,946 | 156 | 12 |
| **SLEEPEEG** | 296,700 | 74,100 | 178 | 5 |
| **ECG** | 1,225,416 | 306,355 | 252 | 4 |

We train all models with a batch size of 8 from 500 epochs on an NVIDIA V100 GPU.

## D DATA SETUP FOR CLUSTERING EVALUATION AND VISUALIZATION

To train our model, we use three public datasets: HARTH, ECG, and SLEEPEEG. We preprocess these datasets using different window and hop lengths in the STFT. The HARTH dataset is processed to have a sequence length of 119 instances (each instance is 1 second, but with half a second overlap during preprocessing via STFT), corresponding to 60 seconds on the original signal, with a feature dimension of 156. The ECG dataset, on the other hand, has a sequence length of 500 instances, corresponding to 250 seconds, with a feature dimension of 252. Lastly, the SLEEPEEG dataset has a sequence length of 300 instances, with each instance representing 200 window size of the signal, and a feature dimension of 178. For model evaluation on clustering, we set aside a balanced subset of all three datasets, not used during the training of the unsupervised representation learning model. Specifically, for the HARTH, we randomly select 1000 instances of all classes, except class 11 (insufficient samples), where we select 500 random samples. Similarly, for the SLEEPEEG, we select random 1000 instances from all classes. Finally, for the ECG, we select random 1% of both majority classes and the entirety of the remaining classes to give a distribution of 1577, 972, 459, and 1467 for all four classes, respectively.

