# OpenReview forum: "Dynamic Contrastive Learning for Time Series Representation"
_ICLR.cc/2025/Conference — Submitted to ICLR 2025_

### Official Review · Reviewer_9T4J · 2024-10-26

**Soundness:** 1
**Presentation:** 2
**Contribution:** 1
**Rating:** 3
**Confidence:** 5

**Summary:**

This paper proposes a dynamic contrastive learning method for time series representation learning, which harness every time step in a sequence as positive and negative pairs. Experiments are carried out on clustering and classification tasks.

**Strengths:**

1. The motivation of introducing SSL for time series modeling is clear.
1. The paper is well-structed.

**Weaknesses:**

1. Related works are not sucfficient. More relevant studies should be discussed, e.g., soft contrastive learning for time series (ICLR 2024), Towards Enhancing Time Series Contrastive Learning: A Dynamic Bad Pair Mining Approach (ICLR 2024)

2. Continue with 1, the contribution and novelty of the proposed approach is unclear. As there are numuous methods that leverage contrastive learning for time series, what is the specific innovations of the proposed approach in comparison with them? I have not seen new insights and new techniques.

3. The experiments are weak. On the one hand, recent advanced baselines, e.g., SimMTM, and those menthoned in #1, are missed. On the other hand, more datasets (e.g., UCR classification datasets) and tasks (e.g., forecasting) should be evaluated.

**Questions:**

Please address the weakness.

---

> ### Author Response · Authors · 2024-11-21
> **Official Response by the Authors to Reviewer 9T4J (1/2)**
>
> We appreciate your valuable feedback and have addressed the weaknesses and questions below.
>
> ---
>
> ### W1: Mention of SoftCLT and Missing Related Work
>
> **Reply:** We mentioned SoftCLT in our related works. Specifically in lines 96-97:
> *"Lee et al. (2024) propose soft assignment to leverage every pair other than the positive pairs by assigning weights to both instance and temporal CL to improve on previous CL frameworks. However, this soft assignment is precomputed offline and not during training."*
>
> We also recognize the unintended omission of [1] and will ensure it is added in the final version.
>
> ---
>
> ### W2: Contributions to Positive Pair Selection and Novelty of MP-Xent
>
> **Reply:** While contrastive learning (CL) in time series (TS) is not a new concept, we revisited positive pair selection to avoid the inductive biases introduced by data augmentation in TS, which are derived from computer vision (CV) practices. Following the direction of TNC, we explored leveraging temporal dimensions for positive pair selection.
>
> Existing TS representation learning approaches using temporal information often rely on inefficient methods for sampling positives. For example:
> - **TNC:** Uses windows to select sub-series of interest, leading to inefficiency as some timestamps are discarded during each batch update.
> - **Our Method:** Uses **N-pair loss** to efficiently utilize every sample in a batch (instance-level), computing an (N+1)-tuple loss during each update.
>
> In summary, our work contributes the following novel additions to TS and CL frameworks:
> 1. Introducing **N-pair loss** for TS CL.
> 2. Extending this loss to accommodate **multiple positives (MP-Xent)**.
>
> These innovations explain why our model achieves the fastest training time among all baselines, as shown in **Table 4.5**.
>
> ---
>
> ### W3: Comparison with Baselines and Context for SoftCLT and Dynamic Bad Pair Mining
>
> **Reply:** While it is ideal to compare with as many baselines as possible, both **SoftCLT** and the **Dynamic Bad Pair Mining Approach (ICLR 2024)** are not standalone CL frameworks. They are designed to enhance existing algorithms.
>
> As stated in our responses to Reviewer MKT3 and f9iC, **SoftCLT** is a "plug-and-play" method that augments existing CL approaches to improve performance. The best results from SoftCLT are achieved by combining it with **TS2Vec** [2], which we have already compared against. Including SoftCLT in our framework would complicate experiments, as it would necessitate adding SoftCLT to all baselines, including our **DynaCL**, for fairness. This would lead to a study on which method benefits the most from SoftCLT, which deviates from the primary focus of this research.
>
> Regarding **SimMTM**, we have conducted experiments on all three datasets, and the results are shown below:
>
> | **Dataset**| **Model** | **Accuracy (%)** | **F1 Score** | **Precision** | **Recall** | **Training Time (s)** |
> |-----------|-----------|--------------|-----------------|----------------|------------|----------------|
> | **HARTH**  | SimMTM    |  30.43 | 0.16 | 0.13 | 0.09            | 19.7k           |
> |   | DynaCL    | **35.06**     | **0.27**           | **0.15**        | **0.12**   | **0.34k**        |
> |   |     |      |           |        |    |         |
> | **SleepEEG**  | SimMTM      |  49.11 | 0.35 | 0.19 | 0.22        | 41.1k            |
> |   | DynaCL    | **57.55**     | **0.56**           | **0.42**        | **0.41**   | **0.06k**        |
> |   |     |      |           |        |    |         |
> | **ECG**  | SimMTM      |  49.94 | 0.39 | 0.28 | 0.26        | 23.8k            |
> |   | DynaCL    | **59.48**     | **0.56**           | **0.33**        | **0.30**   | **0.19k**        |
>
>
> *Note:* We trained for 50 epochs and used a single seed to enable us to get the results on time for the rebuttal.

---

> > ### Author Response · Authors · 2024-11-21
> > **Official Response by the Authors to Reviewer 9T4J (2/2)**
> >
> > ### Additional Datasets and Tasks
> >
> > **Reply:** Our methodology is based on the assumption that **similarity exists within nearby instances**:
> > *"Consecutive instances in a sequence share the same class, and event labels do not change too often."*
> >
> > This condition is prevalent in real-world TS datasets with repeated labels along the temporal dimension, which is why we used such datasets, following conventions set by **[3], [4], [5], and [6]**. In contrast, datasets from archives like **UCR** or **UEA** often do not meet this criterion.
> >
> >
> > While we agree that including more tasks would be beneficial, **time series forecasting** is outside the scope of this work. For example, **CoST** [7] focuses mainly on forecasting without requiring classification tasks. Nonetheless, we appreciate your suggestions.
> >
> >
> > **Thank you again for your valuable feedback!**
> >
> >
> > [1] Towards Enhancing Time Series Contrastive Learning: A Dynamic Bad Pair Mining Approach, ICLR 2024 \
> > [2] TS2Vec: Towards Universal Representation of Time Series, AAAI 2022
> > [3] TF-C: Self-Supervised Contrastive Pre-Training For Time Series via Time-Frequency Consistency, Neurips 2022 \
> > [4] TNC: Unsupervised Representation Learning for Time Series with Temporal Neighborhood Coding, ICLR 2021 \
> > [5] TimeDRL: Disentangled Representation Learning for Multivariate Time-Series, ICDE 2024 \
> > [6] TS-TCC: Time-Series Representation Learning via Temporal and Contextual Contrasting, IJCAI 2021 \
> > [7] CoST: Contrastive Learning of Disentangled Seasonal-Trend Representations for Time Series Forecasting, ICLR 2022

---

> > > ### Comment · Reviewer_9T4J · 2024-11-25
> > >
> > > Thanks to the authors for the rebuttal. Considering the technical quality and novelty, I'll maintain my current score.

---

### Official Review · Reviewer_vi2d · 2024-10-27

**Soundness:** 2
**Presentation:** 2
**Contribution:** 2
**Rating:** 3
**Confidence:** 4

**Summary:**

In this paper, the authors present  Dynamic Contrastive Learning (DynaCL), a new approach for unsupervised representation learning in time series analysis. DynaCL employs contrastive learning by pairing adjacent time steps as positives, leveraging an N-pair loss strategy to treat all samples within a batch as potential positive or negative pairs. Experimental results demonstrate that DynaCL embeds instances from time series into semantically meaningful clusters, allowing superior performance on downstream tasks on various public time series datasets.

**Strengths:**

1. The authors compare **DynaCL** against state-of-the-art baselines for time series representation learning on clustering quality and event classification, these are uncommon but meaningful in the time series representation domain.
2.  The notation used by the authors in the paper is relatively consistent and clear.
3. The authors' experiments are relatively comprehensive.

**Weaknesses:**

Generally, the technique of the manuscript is partially sound, however, the novelty is marginal and some major concerns are listed below:
1. There is a significant logical issue in this paper: the authors claim that the main focus of contrastive learning for time series lies in the construction of positive and negative sample pairs. However, contrastive learning generally involves three key components: constructing positive and negative pairs, designing an encoder to map the original data to a hidden space, and formulating the contrastive loss function [1]. Thus, the authors do not necessarily find the most critical step but rather assert that sample pair construction is the most important. Furthermore, it is evident throughout the paper that the design or selection of the contrastive loss function is also essential, especially in the related work section. This weakens the clarity of the paper’s motivation.
2. The authors aim to convey that "understanding events in time series is an important task in various contexts." However, the paper's self-supervised learning approach focuses on moments, and it lacks an adequate explanation to help readers understand how modeling moments contributes to understanding events. Additionally, the authors need some references in the opening sentence of the introduction on line 28 to support the prevalence of event labeling.
3. The authors claim that “Currently, TS representation learning that leverages temporal information in the contrastive objective relies on complicated statistical approaches for sampling positives”. However, there are many existing methods for constructing positive and negative sample pairs, not limited to complex statistical approaches. The references cited by the authors are not very up-to-date, with the most recent being IJCAI 2023.
4. The innovation of this paper is modest, as the emphasized addition of positive samples can be viewed as a special case of [2].

[1] Ts2vec: Towards universal representation of time series, AAAI 2022.

[2] Unsupervised Representation Learning for Time Series with Temporal Neighborhood Coding, ICLR 2021.

**Questions:**

Please answer the following questions combined with the content in weakness.

Why does DynaCL-M still fail to perform consistently well across the two downstream tasks, even after adding two augmentations to DynaCL?

---

> ### Author Response · Authors · 2024-11-21
> **Official Response by the Authors to Reviewer vi2d**
>
> We thank you for the effort and time spent providing such detailed and valuable feedback. Your sentence-level critique is particularly appreciated and will help us refine our phrasing to ensure clarity and avoid potential misunderstandings.
>
> ---
>
> ### W1: Positive and Negative Pair Selection, Encoder Design, and Loss Function
>
> **Reply:** Thank you for highlighting this point. Contrastive learning (CL) frameworks inherently involve three key components: selecting positive and negative pairs, designing an encoder, and formulating the contrastive loss function. Our primary contribution is introducing a contrastive loss function that effectively captures temporally adjacent instances as positive pairs, thereby unifying two critical steps in the CL pipeline.
>
> For encoder design, we maintain fairness by using the same encoder network across all baselines. We will clarify this distinction in the updated version of our manuscript.
>
> ---
>
> ### W2: Clarifying the Core Idea and Adding References
>
> **Reply:** We will revise the sentence to better communicate our paper's central idea: learning meaningful representations through an unsupervised contrastive learning framework. Additionally, we will incorporate the suggested references to strengthen the context.
>
> ---
>
> ### W3: Positives and Negatives Sampling
>
> **Reply:** Regarding the statement about existing methods using "complicated statistical approaches for sampling positives," we have revised this to a more modest tone: "inefficient approaches for sampling positives." We welcome suggestions regarding the specific methods you mentioned for constructing positive and negative sample pairs. Incorporating these insights will enable us to refine the related works section and evaluate whether these methods can achieve the same efficiency in batch updates as our N-pair loss framework.
>
> ---
>
> ### W4: Differences from TNC
>
> **Reply:** Our approach differs from TNC [1] in the following key aspects:
>
> 1. **Focus on Instances:**
>    TNC focuses on timestamps within a single instance, while our method focuses on entire instances within longer sequences.
>
> 2. **Sampling Efficiency:**
>    TNC uses sliding windows to select subsequences of interest, which can lead to inefficient sampling as some timestamps are discarded in each batch update. In contrast, we employ N-pair loss to utilize every sample in a batch for computing an (N+1)-tuple loss in each update step.
>
> 3. **Positive and Negative Pair Selection:**
>    TNC selects a single positive pair and a single negative pair, akin to Triplet Loss [2]. Our method selects multiple positive pairs and considers all other instances in the batch as negatives, similar to [3].
>
> 4. **Sampling Approach:**
>    TNC uses Augmented Dickey-Fuller (ADF) statistical test to select regions of interest for positives. We simplify this process by selecting only temporally adjacent instances as positives.
>
> Our main contribution is the introduction of N-pair loss to time series contrastive learning and its extension to support multiple positives (MP-Xent). This innovation explains why our model achieves the fastest training time among all baselines, as demonstrated in Table 4.5.
>
> ---
>
> ### Q1: DynaCL-M vs. DynaCL
>
> **Reply:** Our primary contribution is the vanilla DynaCL model. We introduced augmentations in DynaCL-M to ensure it outperformed TNC in clustering, as TNC used clustering as an evaluation metric.
>
> The observation that DynaCL-M performs worse on downstream tasks than vanilla DynaCL challenges the notion that achieving high scores on unsupervised clustering metrics does not necessarily imply that the learned embeddings are meaningful or effective in downstream tasks. Investigating why DynaCL-M fails to achieve consistent performance across all downstream tasks is a priority for our future work.
>
>
> [1] TNC: Unsupervised Representation Learning for Time Series with Temporal Neighborhood Coding, ICLR 2021 \
> [2] T-Loss: Unsupervised scalable representation learning for multivariate time series, Neurips 2019 \
> [3] SimCLR: A Simple Framework for Contrastive Learning of Visual Representations, ICML 2020

---

> > ### Comment · Reviewer_vi2d · 2024-11-25
> >
> > I have read the authors' responses. While it does not fully address my concerns, I appreciate the effort they have put into it. However, there is still room for improvement in the paper, such as clarifying the motivation and refining certain expressions. Additionally, the modeling approach really shares some similarities with existing methods. Therefore, I still recommend that the authors make substantial revisions in the future.

---

### Official Review · Reviewer_f9iC · 2024-10-31

**Soundness:** 3
**Presentation:** 3
**Contribution:** 3
**Rating:** 5
**Confidence:** 4

**Summary:**

This paper introduces Dynamic Contrastive Learning (DynaCL), an unsupervised method for embedding time series data to improve performance in subsequent classification and detection tasks with minimal human intervention. DynaCL defines positive pairs based on temporal adjacency, utilizing N-pair loss to dynamically adjust sample pairings during training. By doing so, it efficiently learns representations that form semantically meaningful clusters, achieving strong performance on various public time series datasets. The study highlights that high unsupervised clustering metrics do not necessarily translate to improved performance in downstream tasks, emphasizing the need for careful consideration of representation quality beyond clustering alone.

**Strengths:**

1. The methodology is clear and the problem is well-motivated.
2. The proposed method has fairly good novelty.
3. The experiments for different setups are detailed.

**Weaknesses:**

1. The author needs to compare against recent works on SOTA time series contrastive learning frameworks like InfoMin, SoftCLT, etc.
2. The author could consider other time-series domains like UCIHAR, UEA, and UCR datasets to evaluate against.

**Questions:**

1. How does the proposed time series representation method perform on sequential tasks, such as the commonly used prediction task?

---

> ### Author Response · Authors · 2024-11-21
> **Official Response by the Authors to Reviewer f9iC**
>
> We thank Reviewer f9iC31 for taking the time to carefully review our work and for providing valuable comments. Below, we address the concerns raised.
>
> ---
>
> ### W1: Comparison with SoftCLT and InfoMin
>
> **Reply:** We acknowledge that SoftCLT is a recent work on contrastive learning (CL) for time series representation submitted to this venue. However, as mentioned in our response to Reviewer MKT3, SoftCLT is not a standalone CL framework. It is designed to enhance existing CL methods. Consequently, there is no direct way to compare our work with SoftCLT alone.
>
> The key results from the SoftCLT paper come from applying it to frameworks like TS2Vec and TNC, both of which we compare against in our work. A direct comparison between a CL framework and SoftCLT would require adding SoftCLT to one of our baselines, which introduces potential biases. To maintain fairness, this would necessitate applying SoftCLT to all other baselines, including our DynaCL, transforming the focus of the study into determining which framework benefits the most from SoftCLT. This deviates from the primary objective of our research.
>
> Regarding InfoMin, we could not locate a time series paper using this framework. If you could provide a reference, we would be happy to evaluate it in future experiments.
>
> ---
>
> ### W2: Evaluation on UCIHAR, UEA, and UCR Datasets
>
> **Reply:** Thank you for this suggestion. While it would indeed be useful to test our method’s generalizability on datasets like UCIHAR, UEA, and UCR, our proposed method assumes the following:
> *"Similarity exists within nearby instances — that consecutive instances in a sequence have the same class, and event labels do not change frequently."*
>
> This condition often holds in real-world time series datasets, where labels are frequently repeated in the temporal dimension. However, it is less applicable to datasets in the UEA or UCR archives. Therefore, we followed conventions established in works such as [1], [2], [3], and [4] by using real-world datasets with long instances.
>
> ---
>
> ### Q1: Prediction Task Evaluation
>
> **Reply:** We agree that evaluating our learned representations in a prediction task would be a valuable addition to this study. We will explore this direction in future work. We sincerely appreciate this suggestion.
>
>
> [1] TF-C: Self-Supervised Contrastive Pre-Training For Time Series via Time-Frequency Consistency, Neurips 2022 \
> [2] TNC: Unsupervised Representation Learning for Time Series with Temporal Neighborhood Coding, ICLR 2021 \
> [3] TimeDRL: Disentangled Representation Learning for Multivariate Time-Series, ICDE 2024 \
> [4] TS-TCC: Time-Series Representation Learning via Temporal and Contextual Contrasting, IJCAI 2021

---

### Official Review · Reviewer_MKT3 · 2024-11-04

**Soundness:** 3
**Presentation:** 3
**Contribution:** 2
**Rating:** 5
**Confidence:** 5

**Summary:**

This paper introduces DynaCL, a novel framework for learning time series representations in an unsupervised manner by leveraging adjacent time steps as positive pairs, thus simplifying the learning process without complex augmentations. Using an MP-Xent loss function, DynaCL effectively captures temporal patterns, enabling high-quality embeddings. An extended version, DynaCL-M, incorporates feature prediction and dynamic margins to enhance clustering, though this sometimes sacrifices downstream task performance. Tested on multiple datasets, DynaCL achieves competitive results in classification and clustering, demonstrating efficient training and a practical approach for time series data representation.

**Strengths:**

1. This paper provides an insightful analysis by contrasting clustering scores (like Davies-Bouldin Index and Silhouette Score) with downstream task performance. This comparison reveals that high clustering metrics do not always equate to useful representations in practical tasks, highlighting the limitations of relying solely on clustering for evaluating representation quality.

2. The introduction of a margin in DynaCL-M is a valuable addition. It allows the model to adjust the separation between positive and negative pairs adaptively based on their temporal proximity, enhancing the flexibility and performance of the learned representations, especially in complex time series where adjacent steps may differ.

**Weaknesses:**

1. **Lack of Novelty in MP-Xent**: The MP-Xent loss function in this paper closely resembles existing approaches like Soft-Nearest Neighbors (SoftCLT), which also leverages multiple positive pairs within a batch. In particular, MP-Xent appears quite similar to soft temporal contrastive learning. This similarity raises questions about the novelty of the loss function, as MP-Xent seems more like an adaptation of established methods than an entirely new approach. Furthermore, MP-Xent can be seen as a specific variant of soft temporal contrastive learning, focused exclusively on temporally adjacent samples, which may limit its generalizability to broader instance-wise contrastive learning contexts.

2. **Limited Datasets in Experimental Evaluation**: The paper’s experiments are limited to just three datasets, which restricts the generalizability of its findings. For a more robust evaluation, the model should be tested on a wider range of datasets, such as the 125 UCR and 29 UEA benchmarks commonly used in time series classification. Additionally, while there is an extensive description of the datasets used, more impactful would be to include semi-supervised and transfer learning experiments, as these are standard benchmarks in time series representation learning and would better demonstrate the model’s versatility across diverse applications.

3. **Missing Important Baselines**: Some key baselines are missing in the comparison, specifically SoftCLT, which has significant conceptual overlap with MP-Xent, and TimeDRL, which also tackles positive pair selection without explicit augmentations (using dropout instead). Excluding them from the experiments limits the comprehensiveness of the baseline comparisons, leaving the results less conclusive than they might otherwise be.

**Questions:**

1. In formula (3), it is evident that both instance-wise and temporal contrastive learning are considered. However, the paper predominantly emphasizes temporal contrastive learning throughout, which creates some confusion. In contrast, SoftCLT clearly differentiates between instance-wise and temporal contrastive learning. Furthermore, does the “multiple positive pairs” mechanism only apply to temporal contrastive learning? If I’m mistaken, please clarify, but it seems this mechanism is not applied to instance-wise contrastive learning. To enhance clarity, it would be beneficial if the authors could further elaborate on their approach to instance-wise versus temporal contrastive learning and how it compares to SoftCLT's handling of these two forms of contrast. This clarification could help address the potential confusion and provide readers with a clearer understanding of the framework's scope and application.
2. In Section 4.5, do all self-supervised learning models use the same backbone architecture? It would be more fair and consistent to use the same backbone encoder across all models for accurate comparison.

---

> ### Author Response · Authors · 2024-11-21
> **Official Response by the Authors to Reviewer MKT3 (1/2)**
>
> We appreciate the thoughtful and insightful comments by Reviewer MKT3. Below are our responses to the weaknesses and questions; we hope this will provide some intuition for the principles underlying our work.
>
> ### W1: Lack of Novelty in MP-Xent
>
> **Reply:** The MP-Xent loss function does appear similar to a specific variant of SoftCLT, as both have a hard assignment on the temporal adjacent steps. However, in practice, this is not the case. The main difference between our model and the approach introduced in SoftCLT lies in how these assignments are computed.
>
> A naive way to select these multiple positives would involve slicing the temporal adjacent instances for all instances in a batch $N$, resulting in a time complexity of $O(N)$, or using SoftCLT's approach to select adjacent instances by looping through the batch $N$ twice to compute an $N \times N$ matrix with soft assignments, resulting in a time complexity of $O(N^2)$. In contrast, our approach combines the positive selection and computation of the MP-Xent loss in constant time complexity $O(1)$ by adapting the N-pair loss technique introduced in [1] and also used in [2]. $N$-pair loss uses every sample in a batch to compute an $N$+1-tuple loss.
>
> Not only is our work the first among all baselines we compared against to adapt the N-pair loss for time series, but we are also the first to extend the N-pair loss to accept multiple positives, which is a non-trivial contribution. This extension enables efficient training, as shown in Table 4.5.
>
> #### Further Explanation of the N-pair Loss
>
> **Vanilla N-pair Loss (Single Positives):**
> 1. Given a batch $N$ of time series instances, create an $N \times N$ similarity matrix by vector multiplication ( $N \times N^T$).
> 2. The lower diagonal elements represent positive pairs (numerator).
> 3. The denominator is the sum of the column vectors minus the positives.
> 4. Use the numerator and denominator to compute the $N$-tuple loss (Equation 1).
>
> **N-pair Loss Extension (Multiple Positives):**
> 1. Given a batch $N$ of time series instances, create an $N \times N$ similarity matrix by vector multiplication ( $N \times N^T$).
> 2. Select the lower diagonal elements.
> 3. The positive pairs are the sum of the shifted left and right of the lower diagonal elements (numerator).
> 4. The denominator is the sum of all elements in the similarity matrix (except the last two columns along each row).
> 5. Subtract a combination of two lower diagonal slices from the sum to produce the denominator (Equation 3).
> 6. Use the numerator and denominator to compute the $N$-tuple loss (Equation 3).
>
> For clarity, we have uploaded the code for our MP-Xent loss (Equation 3) as part of the additional file.
>
> ---
>
> ### W2: Limited Datasets in Experimental Evaluation
>
> **Reply:** Thank you for this suggestion and for providing a specific dataset to test our method's generalizability. Our method rests on the fundamental assumption:
> *"We assume similarity within nearby instances — that consecutive instances in a sequence have the same class, and event labels do not change too often."*
> This condition typically holds for real-world datasets but generally does not for datasets in the UCR or UEA archives. Therefore, we follow conventions in works like [3], [4], [5], and [6] by using long instances of real-world datasets.
>
> We agree that adding semi-supervised and transfer learning experiments would strengthen the evaluation. However, we followed precedence from TNC, and we thought our findings on the clustering task would be more beneficial to report this to the community than including additional tasks.  Additionally, we included t-SNE visualizations of the learned embeddings, which are typically absent in TS contrastive learning papers, to demonstrate the meaningfulness of our representations.

---

> > ### Author Response · Authors · 2024-11-21
> > **Official Response by the Authors to Reviewer MKT3 (2/2)**
> >
> > ### W3: Missing Important Baselines
> >
> > **Reply:** SoftCLT is not a standalone contrastive learning framework. In the authors' words, it is a "plug-and-play" method built on top of existing contrastive learning approaches to improve performance. The best result from the paper combines TS2Vec + SoftCLT. Since we already compared against TS2Vec [7], adding SoftCLT to this framework would require applying it to all other baselines (including DynaCL) for fairness. This would shift the focus of our work to evaluating which method benefits most from SoftCLT, deviating from our main research objectives.
> >
> > Thank you for suggesting the TimeDRL baseline. As it is a recent and contemporary work, we initially missed it. We have now conducted experiments on TimeDRL and CoST [8] (another recent work from this same venue). The results are presented in the table below:
> >
> > | **Dataset**| **Model** | **Accuracy (%)** | **F1 Score** | **Precision** | **Recall** | **Training Time (s)** |
> > |-----------|-----------|--------------|-----------------|----------------|------------|----------------|
> > |   | TimeDRL   | 32.02±1.47          | 0.19±0.02             | 0.17±0.03            | 0.12±0.03        | 6.8k            |
> > | **HARTH**  | CoST    |  32.74±9.67 | 0.26±0.06 | 0.17±0.02 | **0.15±0.02**            | 13.4k           |
> > |   | DynaCL    | **37.95±4.51**     | **0.29±0.06**           | **0.18±0.04**        | 0.13±0.02   | **3.4k**        |
> > |   |     |      |           |        |    |         |
> > |   | TimeDRL   | 48.77±0.43          | 0.34±0.00             | 0.24±0.03            | 0.21±0.00        | 0.9k            |
> > | **SleepEEG**  | CoST      |  50.30±0.23 | 0.39±0.01 | 0.32±0.00 | 0.26±0.01        | 1.5k            |
> > |   | DynaCL    | **62.08±0.64**     | **0.60±0.01**           | **0.52±0.01**        | **0.50±0.01**   | **0.6k**        |
> > |   |     |      |           |        |    |         |
> > |   | TimeDRL   | 51.08±2.36          | 0.44±0.04             | 0.28±0.02            | 0.26±0.01        | 8.9k            |
> > | **ECG**  | CoST      |  55.82±4.95 | 0.50±0.08 | 0.31±0.03 | 0.29±0.02        | 3.5k            |
> > |   | DynaCL    | **58.74±0.62**     | **0.56±0.01**           | **0.32±0.00**        | **0.30±0.00**   | **1.9k**        |
> >
> >
> >
> > Our DynaCL model achieves better accuracy and trains faster than both models using the same backbone architecture.  We intend to include these results in any future versions of this paper.
> >
> > ---
> >
> > ### Q1: Formula (3)
> >
> > We understand the confusion and will enhance clarity in the final version of our paper. Our method handles time series at the **instance level** rather than the **timestamp level**. Each TS dataset contains a series of instances, and these consecutive instances are stacked together to form the entire temporal dimension in a long sequence.
> >
> > When we refer to contrasting in the temporal dimension, we mean adjacent instances within this sequence (not adjacent timestamps as in SoftCLT temporal contrastive). In Equation 3, the index $i$ refers to a specific batch, and $t$ denotes an instance within that batch, with $t-1$ and $t+1$ referring to adjacent instances.
> >
> > ---
> >
> > ### Q2: Section 4.5
> >
> > Yes, as stated in Lines 351-353 of the paper:
> > *"Additionally, to eliminate any performance differences arising from variations in model architecture, we use the same encoder network across all baselines. We aim to compare the learning frameworks independent of the choice of encoder."*
> >
> >
> > [1] Improved deep metric learning with multi-class n-pair loss objective, Neurips 2016 \
> > [2] SimCLR: A Simple Framework for Contrastive Learning of Visual Representations, ICML 2020 \
> > [3] TF-C: Self-Supervised Contrastive Pre-Training For Time Series via Time-Frequency Consistency, Neurips 2022 \
> > [4] TNC: Unsupervised Representation Learning for Time Series with Temporal Neighborhood Coding, ICLR 2021 \
> > [5] TimeDRL: Disentangled Representation Learning for Multivariate Time-Series, ICDE 2024 \
> > [6] TS-TCC: Time-Series Representation Learning via Temporal and Contextual Contrasting, IJCAI 2021 \
> > [7] TS2Vec: Towards Universal Representation of Time Series, AAAI 2022 \
> > [8] CoST: Contrastive Learning of Disentangled Seasonal-Trend Representations for Time Series Forecasting, ICLR 2022

---

> > > ### Comment · Reviewer_MKT3 · 2024-11-25
> > >
> > > W1's response:
> > > I now better understand the differences between MP-Xent and SoftCLT, particularly regarding the computational efficiency and the extension to multiple positives. However, I believe it would be helpful to make this distinction more explicit in the paper, especially in terms of the similarities and differences compared to SoftCLT. This would clarify the novelty of your approach and address potential concerns about overlap for readers.
> > >
> > > W2's response:
> > > While I understand the reliance on real-world datasets due to the assumption of temporal similarity, this reasoning should be explicitly stated in the paper for clarity. Readers might otherwise question the exclusion of standard benchmarks like UCR and UEA.
> > > Regarding the t-SNE visualizations and clustering results, while they offer some insights, they are insufficient to conclude that the representations are well-learned. Just as clustering metrics alone cannot fully capture the quality of learned representations, the same critique applies to t-SNE visualizations. In fact, clustering metrics at least provide quantifiable results, whereas t-SNE offers only a subjective, visual assessment without numerical rigor.
> > > Finally, while I understand following TNC’s precedence, semi-supervised and transfer learning experiments are critical for demonstrating real-world applicability, as they reflect primary use cases for self-supervised learning. Including these would significantly strengthen the paper’s impact.
> > >
> > > W3's response:
> > > While SoftCLT is a "plug-and-play" method, its combination with TS2Vec is noted as the best-performing setup in its paper. Given the conceptual overlap between SoftCLT and your approach, comparing DynaCL directly to TS2Vec + SoftCLT would provide valuable context and strengthen your evaluation. Also, in W1, you mentioned that the MP-Xent loss has a time complexity of $O(1)$ compared to SoftCLT's $O(N^2)$. It would be insightful to include a comparison of training times between DynaCL and TS2Vec + SoftCLT to highlight this advantage.

---

> > > > ### Author Response · Authors · 2024-12-01
> > > > **Official Response by the Authors to Reviewer**
> > > >
> > > > Thank you for the invaluable and constructive comments on our work. We are grateful for the time and effort you have put into this review process to help us improve this work. We have noted all your suggestions and feedback and will take this as a direction for our future work.

---

### Meta-Review · Area_Chair_1y7x · 2024-12-21

**Metareview:**

This paper introduces DynaCL, an unsupervised framework for learning time series representations by leveraging adjacent time steps as positive pairs. A key contribution claimed by the authors is the multiple positive cross-entropy (MP-Xent) loss function. However, the MP-Xent loss exhibits notable similarities to existing methods, such as SoftCLT and temporal contrastive learning approaches, which limits its novelty. Comparisons with several recent state-of-the-art baselines published in 2023 and 2024 were not included, making it difficult to evaluate DynaCL’s relative performance comprehensively. Although the current evaluation on three datasets shows promising results, expanding the benchmarks to include broader datasets such as UCR and UEA would better show the robustness and generalizability of the proposed method and further enhance the paper. For these reasons, I recommend rejecting this paper.

**Additional Comments On Reviewer Discussion:**

During the rebuttal period, three out of four reviewers responded to the authors' replies. Reviewer MKT3 still had the concerns about the evaluation, specifically on 1) the comparison with TS2Vec+SoftCLT and  2) semi-supervised and transfer learning scenarioes.  Reviewer vi2d stated that the rebuttal did not fully address the concerns regarding the novelty and motivation of the work, suggesting that substantial revisions are required. Reviewer 9T4J maintained his/her score considering that the technical quality and novelty are insufficient.

---

### Decision · Program_Chairs · 2025-01-22

Reject